# NICU Admissions for Meconium Aspiration Syndrome before and after a National Resuscitation Program Suctioning Guideline Change

**DOI:** 10.3390/children6050068

**Published:** 2019-05-07

**Authors:** Erika M. Edwards, Satyan Lakshminrusimha, Danielle E. Y. Ehret, Jeffrey D. Horbar

**Affiliations:** 1Vermont Oxford Network, Burlington, VT 05401, USA; dehret@vtoxford.org (D.E.Y.E.); horbar@vtoxford.org (J.D.H.); 2Department of Pediatrics, Robert Larner, MD College of Medicine, University of Vermont, Burlington, VT 05405, USA; 3Department of Mathematics and Statistics, College of Engineering and Mathematical Sciences, University of Vermont, Burlington, VT 05405, USA; 4Department of Pediatrics, UC Davis Health, Sacramento, CA 95917, USA; slakshmi@ucdavis.edu

**Keywords:** meconium aspiration syndrome, meconium-stained amniotic fluid, neonatal intensive care unit, Neonatal Resuscitation Program, endotracheal suction

## Abstract

The Textbook of Neonatal Resuscitation, seventh edition, does not suggest routine endotracheal suctioning for non-vigorous infants born through meconium-stained amniotic fluid. We compared 301,150 infants at ≥35 weeks’ gestational age inborn at 311 Vermont Oxford Network member centers in the United States (U.S.) and admitted to neonatal intensive care units (NICU) who were born before (2013 to 2015) and after (2017) the guideline change. Logistic regression models adjusting for clustering of infants within centers were used to calculate risk ratios. NICU admissions for infants with a diagnosis of meconium aspiration syndrome (MAS) decreased from 1.8% to 1.5% (risk ratio: 0.82; 95% confidence interval: 0.68, 0.97) and delivery room endotracheal suctioning in this group decreased from 57.0% to 28.9% (0.51; 0.41, 0.62). Treatment with conventional or high frequency ventilation, inhaled nitric oxide, or extracorporeal membrane oxygenation remained unchanged 42.3% vs. 40.3% (0.95; 0.80, 1.13) among infants with MAS and 9.1% vs. 8.2% (0.91; 0.87, 0.95) among infants without MAS. The use of surfactant among infants with MAS increased from 24.6% to 30% (1.22; 1.02, 1.48). Mortality (2.6 to 2.9%, 1.12; 0.74, 1.69) and moderate/severe hypoxic-ischemic encephalopathy (5.4 to 6.8%, 1.24; 0.91, 1.69) increased slightly in 2017. Subgroup analyses of infants with 1 min Apgar scores of ≤3 found similar results. While NICU admissions for MAS and tracheal suctioning decreased after the introduction of the new guideline with no subsequent increase in severe respiratory distress among infants with and without a MAS diagnosis, limitations in our study preclude inferring that the new guideline is safe or effective.

## 1. Introduction

Approximately 10% to 15% of all infants are born through meconium-stained amniotic fluid (MSAF), of whom 3% to 9% develop meconium aspiration syndrome (MAS) [1]. Eight to 20% of infants born through MSAF are depressed and non-vigorous [1,2] with bradycardia, inadequate respiratory effort and poor tone [3].

Every five years, the Neonatal Resuscitation Program (NRP) updates materials to reflect the new American Heart Association/American Academy of Pediatrics (AHA/AAP) neonatal resuscitation guidelines. The 2015 AHA/AAP Guidelines for Neonatal Resuscitation [4] were released and published in the Textbook of Neonatal Resuscitation (seventh edition) in 2016 [5] and adopted by the American College of Obstetricians and Gynecologists in early 2017 [6]. The sixth edition guidelines for resuscitation of non-vigorous infants born through MSAF recommended routine suctioning of the mouth and trachea [7]. The seventh edition guidelines recommended positive pressure ventilation for a non-vigorous infant born through MSAF if the infant was not breathing or the heart rate was less than 100 beats per minute. Routine intubation for tracheal suction was not suggested [4,5,6].

The change was based on an International Liaison Committee on Resuscitation (ILCOR) evaluation that found insufficient evidence to support routine tracheal suctioning in non-vigorous infants born through MSAF [1]. Three small randomized controlled trials in non-vigorous infants born through MSAF (two of which ILCOR deemed to be low-quality evidence [1]) have compared routine suction vs. not routinely suctioning [8,9,10]. The first two studies did not report any differences in the incidence of MAS. However, Singh et al. reported increased MAS in the group without routine suctioning (*p* = 0.05). In addition, a retrospective cohort study of 231 non-vigorous infants born through MSAF at four delivery services in Texas compared neonatal intensive care unit (NICU) admissions among infants ≥35 weeks’ gestation before and after adoption of the seventh edition of the Textbook of Neonatal Resuscitation guidelines [5] and observed a statistically significant increase in admissions for respiratory distress or failure (22% vs. 40%; odds ratio (OR): 2.2; 95% confidence interval (CI): 1.2, 3.9) and an increase in admissions for MAS that was not statistically significant (5% vs. 11%; OR: 2.3; 95% CI: 0.83, 6.2) [2]. Dr. Wiswell, who conducted a large randomized controlled trial of 2094 apparently vigorous infants [3], recently called for large, high-quality randomized controlled trials of intubation and suctioning of non-vigorous infants born through MSAF to inform future decision making.

Given the challenges conducting trials of this nature, data collected as part of a multicenter national neonatal network may contribute to our understanding of suctioning practices and outcomes in a broad group of centers. Our aim was to compare the incidence, outcomes, and interventions of infants ≥35 weeks’ gestation and admitted to NICUs with and without MAS born before and after the guideline change among Vermont Oxford Network member centers in the U.S.

## 2. Materials and Methods

### 2.1. Data Collection

The Vermont Oxford Network is a voluntary worldwide community of practice dedicated to improving the quality, safety and value of care through a coordinated program of data-driven quality improvement, education, and research. Members participating in the Expanded Database submitted de-identified data on all infants admitted to a NICU, defined as any location within the center in which newborn infants received continuous positive airway pressure or intermittent mandatory ventilation not including areas where these modalities of respiratory support were used only for brief periods of stabilization prior to transfer to another location (Appendix A). Local staff collected infant data using uniform definitions [11] until death, discharge home, or transfer to other centers. All data underwent automated checks for quality and completeness at the time of submission. The University of Vermont Committee on Human Research determined that the use of the Vermont Oxford Network Research Repository for this analysis was not human subjects research. 

### 2.2. Data Analysis

We evaluated infants ≥35 weeks’ gestational age who were inborn at 311 Vermont Oxford Network U.S. member centers with obstetric services that participated in the Expanded Database. We compared infants admitted to NICUs born before (2013 to 2015) and after (2017) the publication of the seventh edition of the Textbook of Neonatal Resuscitation [5]. 

We assessed maternal and infant characteristics, interventions, and outcomes. Death was defined as occurring during the initial hospitalization period; infants transferred to another center before discharge home were followed until ultimate disposition. Diagnosis of MAS required the following criteria: MSAF at birth; respiratory distress within one hour of birth; PaO_2_ < 50 mmHg, central cyanosis in room air, or a need for supplemental oxygen to maintain PaO_2_ > 50 mmHg; abnormal chest x-ray compatible with the diagnosis of meconium aspiration; and absence of culture-proven early onset bacterial sepsis or pneumonia. Hypoxic-ischemic encephalopathy (HIE) defined as moderate (lethargic or mild stupor) or severe (deep stupor or coma) reflected the worst stage observed during the first seven days after birth. Length of stay was measured as the number of days from the date the infant was admitted until the date of discharge home or death. Among infants with MAS, tracheal suctioning was defined as occurring through an endotracheal tube or suction catheter in the delivery room or initial resuscitation area whether meconium was recovered or not. We conducted subgroup analyses of with 1 min Apgar scores ≤3 with and without MAS. An Apgar score of ≤3 was utilized as a proxy for non-vigorous state to reflect low heart rate, poor respiratory effort, and/or poor tone.

Generalized estimating equation logistic regression models adjusting for clustering of infants within centers were used to calculate risk ratios (RR) and 95% CI with Statistical Analysis Software (SAS), version 9.4 (Cary, NC, USA).

## 3. Results

### 3.1. All Infants

Maternal and infant characteristics for 301,150 infants ≥35 weeks’ gestational age admitted to NICUs at the 311 U.S. centers were largely unchanged from the 2013 to 2015 cohort to 2017 cohort (Table 1).

The proportion of all infants admitted to NICUs diagnosed with MAS decreased from 1.8% from 2013 to 2015 to 1.5% in 2017 (risk ratio (RR): 0.82; 95% CI: 0.68, 0.97); Table 1). Among infants ≥35 weeks’ gestational age who were admitted to NICUs and diagnosed with MAS (Table 2), endotracheal suctioning in the delivery room decreased from 57.0% to 28.9% (RR: 0.51; 95% CI: 0.41, 0.62). Use of surfactant at any time increased from 24.6% to 30.0% (RR: 1.22; 95% CI: 1.02, 1.48). Among infants diagnosed with MAS, the combined incidence of use of conventional or high frequency mechanical ventilation, inhaled nitric oxide, or ECMO remained relatively unchanged at 42.3% to 40.3% (RR: 95% CI: 0.95; 0.80, 1.13). Mortality (2.6 to 2.9%, RR: 1.12; 95% CI: 0.74, 1.69) and moderate/severe hypoxic-ischemic encephalopathy (5.4 to 6.8%, RR: 1.24; 95% CI: 0.91, 1.69) increased slightly in 2017.

Among infants ≥35 weeks’ gestational age who were admitted to NICUs without MAS (Table 3), conventional or high frequency mechanical ventilation, inhaled nitric oxide, or ECMO declined from 9.1% in 2013 to 2015 to 8.2% in 2017 (RR: 0.91; 95%CI: 0.87, 0.95), driven by a decrease in the use of conventional or high frequency mechanical ventilation. There was no significant change in use of surfactant among infants without MAS, 3.8% in 2013 to 2015 to 3.5% in 2017 (RR: 0.93; 95% CI: 0.87, 1.00) nor a significant change in the incidence of moderate to severe HIE, 0.6% in 2013 to 2015 to 0.7% in 2017 (RR: 1.14; 95% CI: 1.00, 1.30).

### 3.2. Infants with 1-Minute Apgar Scores ≤3

In subgroup analyses of infants ≥35 weeks’ gestational age with 1 min Apgar scores ≤3 admitted to NICUs, the proportion of infants diagnosed with MAS decreased from 6.8% in 2013 to 2015 to 4.6% in 2017 (RR: 0. 68; 95% CI: 0.60, 0.78)). Among such infants (Table 4), endotracheal suctioning in the delivery room decreased from 82.4% to 52.1% (RR: 0.63; 95% CI: 0.56, 0.71). The combined measure of treatment with conventional or high frequency mechanical ventilation, inhaled nitric oxide, or ECMO did not change significantly (RR: 1.08; 95% CI: 0.97, 1.20) although use of inhaled nitric oxide increased from 16.2% to 21.9% (RR: 1.35; 95% CI: 1.08, 1.69). Surfactant at any time also increased from 27.7% to 36.0% (RR: 1.30; 95% CI: 1.09, 1.55). There were no significant changes in death or pneumothorax; however, moderate to severe HIE increased from 12.1% to 20.1% (RR: 1.67; 95% CI: 1.27, 2.19). 

Among infants ≥35 weeks’ gestational age with 1 min Apgar scores ≤3 admitted to NICUs without a diagnosis of MAS (Table 5), the proportion of infants diagnosed with moderate to severe HIE increased from 5.5% to 6.8% (RR: 1.25; 95% CI: 1.09, 1.42).

## 4. Discussion

After the Textbook of Neonatal Resuscitation (seventh edition) [5] did not suggest routine suctioning of meconium in non-vigorous infants born through MSAF, the percentage of NICU admissions represented by infants ≥35 weeks’ gestation with a diagnosis of MAS and the percentage of such infants who received tracheal suctioning both decreased among all infants and infants with 1 min Apgar scores of ≤3. 

If treatment with conventional or high frequency mechanical ventilation, inhaled nitric oxide, or ECMO is a marker of severe newborn respiratory distress, the reduction in tracheal suctioning among infants diagnosed with MAS was not associated with an increased use of these interventions. An increase in the use of surfactant and, among infants with 1 min Apgar scores of ≤3, inhaled nitric oxide, may have contributed to this decrease [12,13]. However, the increased use of surfactant and iNO can also be secondary to more severe respiratory morbidity associated with MAS in 2017. In addition, the increase in the incidence of moderate to severe HIE among infants with low Apgar scores with MAS in 2017 can potentially suggest more severe perinatal depression associated with less frequent tracheal suctioning. Among infants who were not diagnosed with MAS, use of conventional or high frequency mechanical ventilation, inhaled nitric oxide, or ECMO decreased, indicating that severe respiratory distress did not increase in these infants. 

We speculate that the increased use of inhaled nitric oxide and surfactant in 2017 for infants with 1 min Apgar scores of ≤3 is partly due to increased dissemination of knowledge regarding early use of these agents to limit progression of hypoxemic respiratory failure and pulmonary hypertension [13]. It is also possible that during 2013 to 2015, non-vigorous infants were intubated and suctioned resulting in a delay in initiation of positive pressure ventilation and Apgar scores remained low at 1 min after birth. In 2017, based on the new guidelines, depressed babies with low heart rate and respiratory effort immediately received positive pressure ventilation. However, the possibility of an association between less frequent tracheal suctioning and severity of MAS (as evidenced by increased use of iNO/surfactant) and incidence of HIE is concerning. 

Infants who continue to have Apgar scores of ≤3 at 1 min despite ventilation represent a sicker cohort and are likely to have more severe perinatal depression. This may be reflected by the reduction in the incidence of low Apgar score ≤3 at 1 min among infants with MAS from 1586/3937 (40.3%) in the 2013 to 2015 cohort to 362/1138 (31.8%) in 2017. Profound perinatal depression and pulmonary hypertension are associated with increased incidence of severe HIE [14]. A higher incidence of moderate to severe HIE and inhaled nitric oxide use in 2017 among infants with MAS and low Apgar scores further supports this speculation. It is also possible that tracheal suctioning might have contributed to lower Apgar score at 1 min in the 2013 to 15 cohort. 

In a retrospective cohort study, Chiruvolu and colleagues found a significant increase in non-vigorous infants born through MSAF admitted to NICUs for respiratory issues after four delivery services in Texas adopted the seventh edition guidelines, and a drop in endotracheal suctioning from 70% to 2% [2]. We included infants ≥35 weeks’ GA to be consistent with Chiruvolu et al. Other observational studies used different gestational age criteria and observed higher rates of MAS. Singh et al. evaluated a large dataset of 415,772 neonates and reported that 1.8% of all neonates and 4.6% of all term NICU admissions had MAS [15]. Using a gestational age cut-off of ≥37 weeks’ at birth, we observed a decline in the incidence of MAS from 2.3% in 2013 to 2015 to 1.7% in 2017 among all infants and from 8.1% in 2013 to 2015 to 5.4% in 2017 among infants with Apgar score of ≤3 at 1 min. Temporal changes, different types of data (administrative vs. clinical), and differences in inclusion criteria (inborn and outborn (transferred) infants vs. inborn infants only) may explain the lower incidence of MAS in our population.

We observed a decrease in non-MAS admissions with severe respiratory distress among all NICU admissions. We do not know which of our infants were non-vigorous, although subgroup analyses of infants with 1 min Apgar scores of ≤3 found results similar to all infants. Finally, we did not observe as dramatic a decrease in tracheal suctioning in 2017. Neonatal resuscitation providers undergoing NRP training during the year 2016 could still be taught using the sixth edition of NRP textbook [7] that recommended intubation and suction of all non-vigorous infants born through MSAF. Their certificate of completion would then be good for two years. Hence, the frequency of tracheal suction among infants diagnosed with MAS in 2017 was still 28.9% among all infants and 52.1% among infants with low Apgar scores. In addition, in our cohort, some infants may have been suctioned in the delivery room and not admitted to the NICU with respiratory distress or failure. 

The American College of Obstetricians and Gynecologists guidelines released in 2014 suggested induction of labor between 41 and 42 weeks and recommended induction of labor at or after 42 weeks [16], which can reduce risk for MAS [17]. The Cochrane review comparing induction of labor at term/post-term pregnancy vs. expectant management showed reduced neonatal mortality. Fewer babies in the labor induction group has MAS compared with a policy of expectant management. (RR 0.50; 95% CI 0.34, 0.73). This guideline may reduce the proportion of infants with MAS. However, the median gestational age in our cohort and the proportion of infants ≥42 weeks did not change between time periods. The 2015 AHA/AAP Guidelines for Neonatal Resuscitation were released in October [4], and the timing was unlikely to have had significant influence on practice in 2015. It was recommended that as of January 1, 2017, all NRP classes must use the seventh edition [18]. Based on a 2-year cycle for NRP certification, it is possible that all providers will not be following the sevenths edition guidelines that recommend against routine suctioning for MSAF in non-vigorous infants until January 2019.

There are several limitations to our study; we do not know the exact date of implementation of revised guidelines at our centers; we do not know the timing of the tracheal suctioning, and whether it occurred before or after use of positive pressure ventilation; we do not know the full denominator of infants with MSAF as our cohort represents only those patients that were admitted to the NICU; we also do not know which infants with MSAF were suctioned, and which infants were non-vigorous although subgroup analyses of infants with 1 min Apgar scores of ≤3 may serve as a good proxy. However, there are no studies correlating non-vigorous infants with a low Apgar score at 1 min. Additionally, we do not know the skill level of the resuscitation providers at the participating centers in our study, and results may differ with varying levels of delivery room expertise. The limitations in our data and observational study design preclude inferring that the new NRP guidelines are either safe or effective. 

## 5. Conclusions

We did not find increases in the incidence or severity of MAS or in severe respiratory distress among infants without MAS after the Textbook of Neonatal Resuscitation (seventh edition) guideline was introduced [5]. However, among infants with low Apgar scores, MAS was more often associated with reduced tracheal suctioning, iNO use, surfactant administration and moderate to severe HIE raising concerns for increased respiratory and neurological morbidity in this cohort. Significant limitations in our study preclude inferring that the new guideline is safe or effective. A large, multicenter, randomized controlled trial evaluating tracheal suction in non-vigorous infants born through MSAF is required. Given the challenges in conducting and interpreting such a trial, a prospective observational study at a wide range of delivery centers may help assess the implementation and impact of the guideline in routine practice.

## Figures and Tables

**Table 1 children-06-00068-t001:** Maternal and infant characteristics among all infants ≥35 weeks’ gestational age at birth.

	2013 to 2015 N = 222,438	2017 N = 78,712
*Maternal Characteristics*		
Race, %		
White non-Hispanic	58.7	56.3
African American	17.3	18.3
Hispanic	15.5	16.1
Other	8.5	9.3
Hypertension, %	17.8	20.6
Chorioamnionitis, %	11.8	11.1
*Infant Characteristics*		
Gestational age, weeks, median (q1, q3)	38 (36, 39)	38 (36, 39)
Birth weight, grams, median (q1, q3)	3120 (2635, 3580)	3118 (2635, 3570)
Male, %	56.5	56.6
Small for gestation, %	18.8	19.0
1 min Apgar score ≤3, %	10.6	10.0
5 min Apgar score ≤7, %	15.0	15.2
Meconium Aspiration Syndrome, %	1.8	1.5

**Table 2 children-06-00068-t002:** Neonatal interventions and outcomes among infants ≥35 weeks’ gestational age admitted to a neonatal intensive care units (NICU) with a diagnosis of meconium aspiration syndrome.

	2013 to 2015 N = 3,937	2017 N = 1,138	RR (95% CI)
*Interventions*			
Endotracheal suctioning, %	57.0	28.9	0.51 (0.41–0.62)
Conventional or high frequency ventilation, inhaled nitric oxide, or ECMO ^a^, %	42.3	40.3	0.95 (0.80–1.13)
Conventional or high frequency ventilation, %	41.8	40.2	0.96 (0.81–1.14)
Inhaled nitric oxide, %	12.1	13.7	1.13 (0.90–1.43)
ECMO, %	1.3	1.1	0.83 (0.41–1.68)
Oxygen at any time, %	92.6	90.0	0.97 (0.91–1.04)
Surfactant at any time, %	24.6	30.0	1.22 (1.01–1.48)
*Outcomes*			
Death, %	2.6	2.9	1.12 (0.74–1.69)
Pneumothorax, %	10.2	8.4	0.82 (0.63–1.08)
Moderate/severe hypoxic-ischemic encephalopathy, %	5.4	6.8	1.24 (0.91–1.69)
Length of stay, days, median (q1, q3)	8 (5, 16)	8 (5, 17)	^b^

^a^ Extracorporeal membrane oxygenation. ^b^ not applicable.

**Table 3 children-06-00068-t003:** Neonatal interventions among all infants ≥35 weeks’ gestational age at birth admitted to a NICU and not diagnosed with meconium aspiration syndrome.

	2013 to 2015 N = 218,475	2017 N = 77,545	RR (95% CI)
*Interventions*			
Conventional or high frequency ventilation, inhaled nitric oxide, or ECMO ^a^, %	9.1	8.2	0.91 (0.87–0.95)
Conventional or high frequency ventilation, %	9.0	8.1	0.91 (0.87–0.95)
Inhaled nitric oxide, %	1.0	1.0	0.99 (0.90–1.09)
ECMO, %	0.3	0.2	0.91 (0.74–1.12)
Oxygen at any time, %	49.4	48.9	0.99 (0.97–1.01)
Surfactant at any time, %	3.8	3.5	0.93 (0.87–1.00)
*Outcomes*			
Death, %	1.1	1.1	1.01 (0.93–1.10)
Pneumothorax, %	3.5	3.3	0.93 (0.88–0.99)
Moderate/severe hypoxic-ischemic encephalopathy, %	0.6	0.7	1.14 (1.00–1.30)
Length of stay, days, median (q1, q3)	6 (4, 11)	6 (4, 11)	^b^

^a^ Extracorporeal membrane oxygenation. ^b^ not applicable.

**Table 4 children-06-00068-t004:** Neonatal interventions and outcomes among infants ≥35 weeks’ gestational age with a 1 min Apgar score ≤3 admitted to a NICU with a diagnosis of meconium aspiration syndrome.

	2013 to 2015 N = 1586	2017 N = 362	RR (95% CI)
*Interventions*			
Endotracheal suctioning, %	82.4	52.1	0.63 (0.56, 0.71)
Conventional or high frequency ventilation, inhaled nitric oxide, or ECMO ^a^, %	57.6	62.2	1.08 (0.97, 1.20)
Conventional or high frequency ventilation, %	57.4	61.9	1.08 (0.97, 1.20)
Inhaled nitric oxide, %	16.2	21.9	1.35 (1.08, 1.69)
ECMO, %	1.8	2.3	1.23 (0.47, 3.19)
Oxygen at any time, %	97.8	98.9	1.01 (0.99, 1.03)
Surfactant at any time, %	27.7	36.0	1.30 (1.09, 1.55)
*Outcomes*			
Death, %	5.3	7.2	1.38 (0.88, 2.16)
Pneumothorax, %	10.3	11.5	1.11 (0.80, 1.55)
Moderate/severe hypoxic-ischemic encephalopathy, %	12.1	20.1	1.67 (1.27, 2.19)
Length of stay, days, median (q1, q3)	9 (5, 17)	10 (6, 20)	^b^

^a^ Extracorporeal membrane oxygenation. ^b^ not applicable.

**Table 5 children-06-00068-t005:** Neonatal interventions among all infants ≥35 weeks’ gestational age at birth with a 1 min Apgar score ≤ 3 admitted to a NICU and not diagnosed with meconium aspiration syndrome.

	2013 to 2015 N = 21,545	2017 N = 7,347	RR (95% CI)
*Interventions*			
Conventional or high frequency ventilation, inhaled nitric oxide, or ECMO ^a^, %	27.2	26.6	0.98 (0.93, 1.03)
Conventional or high frequency ventilation, %	27.1	26.6	0.98 (0.94, 1.03)
Inhaled nitric oxide, %	3.3	3.2	0.98 (0.86, 1.13)
ECMO, %	0.7	0.5	0.70 (0.49, 0.99)
Oxygen at any time, %	90.4	90.4	1.00 (0.99, 1.01)
Surfactant at any time, %	5.4	5.7	0.93 (0.87, 1.00)
*Outcomes*			
Death, %	5.6	5.7	1.02 (0.91, 1.14)
Pneumothorax, %	5.0	5.2	1.03 (0.91, 1.16)
Moderate/severe hypoxic-ischemic encephalopathy, %	5.5	6.8	1.25 (1.09, 1.42)
Length of stay, days, median (q1, q3)	6 (4, 11)	6 (4, 11)	^b^

^a^ Extracorporeal membrane oxygenation. ^b^ not applicable.

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
