# Peer review of "NICU Admissions for Meconium Aspiration Syndrome before and after a National Resuscitation Program Suctioning Guideline Change"

_children, 2019, doi:10.3390/children6050068_

Round 1

Reviewer 1 Report

The manuscript of Edwards, et al. use data from the Vermont Oxford Network (VON) to compare the clinical aspects (including the frequency) of meconium aspiration syndrome (MAS) for the three years (2013-2015) prior to the publication of the 2016Textbook of Neonatal Resuscitation (7thedition) to the subsequent year (2017).  The recommendation was to no longer routinely intubate and suction non-vigorous infants born through meconium-stained amniotic fluid (MSAF).  One has to recognize that the latter recommendation was not based on a randomized controlled trial.  By contrast, the two previous subsequent major changes concerning MSAF (management of vigorous infants and amnioinfusion) were based on large, well-done clinical trials.  In 2018 appeared two investigations, those of Chiruvolu et al. and of Singh et al. (the authors references number 2 and 10) in which the results appeared to indicate that the 2016 change in MSAF recommendation may have been done too hastily…. non-vigorous babies that were no longer being intubated and suctioned had worse outcomes.  In many respects, the current manuscript is a post hocattempt to find data to justify the 2016 change.  Unfortunately, there are major problems with this manuscript.

The authors were unable to specifically identify MSAF infants who were deemed to be non-vigorous in the first 10-15 seconds following birth, nor were they able to precisely determine how such infants were managed and what were their exact outcomes.  The authors make a number of assumptions and speculations about their data.  As a long-time NRP instructor (for many decades), a major concern is that we have no idea how many of the babies born in the 2017 cohort were actually treated using the new recommendations.  People undergoing NRP training during the year 2016 could still be taught using the 6thedition of the NRP textbook that recommended intubating and suctioning all non-vigorous MSAF infants.  Their certificate of completion would then be good for two years.  Hence, there may have been a sizable number of infants in the 2017 cohort managed exactly like those in the 2013-2015 group.  

I believe it would be more representative of effect of the new recommendations if the authors chose to compare data in a period of time in which we know that all clinicians will have undergone training only using the 7thedition guidelines.  This would mean comparing their control group (2013-2015) to babies born after January 1, 2019.   Previous publications have found that there has been an ongoing decrease the frequency of MAS since long before 2013.  Hence, the purported decline in the 2017 MAS incidence may only reflect a continuation of this trend and not be due to the “new” NRP recommendation.  Moreover, some population-based studies have reported sizable differences in MAS frequency from one year to the next.  Thus, it would be considerably better for the authors to compare a population of infants born over a three year period of (eg. 2019-2021), similar to their control group.

It is very unusual to present data as percentages in tables 2-5, rather than giving the exact number followed by the percentage in parentheses.  I request that the authors redo these tables and present it in the more standard fashion. I would also ask the authors to limit the population to babies born >37 weeks.  These are the infants that are substantially more likely to be born through MSAF.  In addition, the VON data indicate that babies with MAS make up approximately 1.5%-1.8% of babies >35 weeks gestation admitted to NICUs.  This is roughly one-third the incidence reported from an even larger population (J Perinatol 2009;29:497-503).  Can the authors comment on this difference? Additionally, can the authors explain whether or not there are efforts by VON to verify data collected at their 311 centers?  This could be done in a small representative sample of sites as a check for accuracy. Otherwise, do we just have to assume that all the data have been collected and entered correctly?

Of interest in this manuscript, among infants with MAS diagnosed in 2017, there were higher incidences of moderate/severe HIE, surfactant use and inhaled nitric oxide use.  This was true in MAS infants overall, as well as in those with one-minute Apgar scores <3. The authors choose to speculate these findings reflect the elimination of less sick infants who respond to ventilation.  What is worrisome is an alternative hypothesis… the lack of intubation and suctioning has led to sicker infants.  Unfortunately, there is no way to be certain whether either of these conjectures is correct. In addition, the authors speculate that the increased use of nitric oxide and surfactant in 2017 is partly due to increased knowledge subsequent to the 2013-2015 epoch about the latter therapies’ benefits in ameliorating hypoxemic respiratory failure.  This is fallacious, as knowledge about such benefits of these therapies have been known for 15-20 years!

The authors should notuse a one-minute Apgar score <3 as a proxy for the non-vigorous MSAF infant.  The status of being non-vigorous has been specifically defined as an infant assessed immediately after birthas having a heart rate < 100 and/or poor respiratory effort and/or poor tone.  Moreover, the intubation procedure in MSAF infants is associated with lower one-minute Apgar scores.   Unless the current investigators can reference a study that has specifically correlated the aforementioned definition of non-vigorous MSAF infants with one-minute Apgar scores, they cannot justify this assumption and should eliminate this category.

I applaud the authors acknowledging the limitations to their study and that this precludes their being able to infer that the new guidelines are safe or effective.  Nevertheless, they stated that their findings are “reassuring”, presumably meaning that the 2016 NRP changes have not resulted in worse outcomes. Unfortunately, I believe the limitations in their study prevent them from saying their results are reassuring, particularly in the face of the 2018 publications of Chiruvolu et al. and Singh et al.  I wholeheartedly concur that a large, multicenter, prospective randomized controlled trial is needed to assess tracheal suction in non-vigorous infants born through MSAF.

Author Response

The manuscript of Edwards, et al. use data from the Vermont Oxford Network (VON) to compare the clinical aspects (including the frequency) of meconium aspiration syndrome (MAS) for the three years (2013-2015) prior to the publication of the 2016Textbook of Neonatal Resuscitation (7thedition) to the subsequent year (2017).  The recommendation was to no longer routinely intubate and suction non-vigorous infants born through meconium-stained amniotic fluid (MSAF).  One has to recognize that the latter recommendation was not based on a randomized controlled trial.  By contrast, the two previous subsequent major changes concerning MSAF (management of vigorous infants and amnioinfusion) were based on large, well-done clinical trials.  In 2018 appeared two investigations, those of Chiruvolu et al. and of Singh et al. (the authors references number 2 and 10) in which the results appeared to indicate that the 2016 change in MSAF recommendation may have been done too hastily…. non-vigorous babies that were no longer being intubated and suctioned had worse outcomes.  In many respects, the current manuscript is a post hoc attempt to find data to justify the 2016 change.  Unfortunately, there are major problems with this manuscript.

Response: We agree with the reviewer that the findings of Chiruvolu et al and Singh et al are concerning. The purpose of this manuscript was to report the response to the change in NRP recommendation (decrease in tracheal suctioning from 57 to 28.9% and 82.4% to 52.1% among infants with low Apgar scores) on the incidence of MAS (1.8 to 1.5%). Interestingly, as pointed out by the reviewer, various markers of severity of MAS have demonstrated an upward trend (use of iNO, surfactant – significant increase; death, moderate/severe HIE – not significant) causing further concern. We have highlighted this aspect further and emphasized the definite need for a multicenter, randomized trial.

The authors were unable to specifically identify MSAF infants who were deemed to be non-vigorous in the first 10-15 seconds following birth, nor were they able to precisely determine how such infants were managed and what were their exact outcomes.  

The reviewer is correct and this is a major limitation of this manuscript.

The authors make a number of assumptions and speculations about their data.  As a long-time NRP instructor (for many decades), a major concern is that we have no idea how many of the babies born in the 2017 cohort were actually treated using the new recommendations.  People undergoing NRP training during the year 2016 could still be taught using the 6thedition of the NRP textbook that recommended intubating and suctioning all non-vigorous MSAF infants.  Their certificate of completion would then be good for two years.  Hence, there may have been a sizable number of infants in the 2017 cohort managed exactly like those in the 2013-2015 group.

We agree that the impact of the change in recommendation may take years. However, the significant reduction in endotracheal suctioning among infants diagnosed with MAS suggests that at least some neonatal providers were probably adapting the new guidelines. We have added the above limitation mentioned by the reviewer to the manuscript.

I believe it would be more representative of effect of the new recommendations if the authors chose to compare data in a period of time in which we know that all clinicians will have undergone training only using the 7thedition guidelines.  This would mean comparing their control group (2013-2015) to babies born after January 1, 2019.   Previous publications have found that there has been an ongoing decrease the frequency of MAS since long before 2013.  Hence, the purported decline in the 2017 MAS incidence may only reflect a continuation of this trend and not be due to the “new” NRP recommendation.  Moreover, some population-based studies have reported sizable differences in MAS frequency from one year to the next.  Thus, it would be considerably better for the authors to compare a population of infants born over a three year period of (eg. 2019-2021), similar to their control group.

We agree with the reviewer and will continue to monitor the incidence of MAS. In the meantime, we would like to alert readers about the impact of reduced tracheal suction on incidence and severity of MAS and associated PPHN and HIE. We intend to conduct a multicenter trial of suctioning vs. no-suctioning for MSAF and non-vigorous term infants. Manuscripts outlining the benefits and risks of not suctioning will be important to generate equipoise for this study.

It is very unusual to present data as percentages in tables 2-5, rather than giving the exact number followed by the percentage in parentheses.  I request that the authors redo these tables and present it in the more standard fashion. 

Given the amount of information in the tables, we felt that the tables looked very cluttered and difficult to read.

I would also ask the authors to limit the population to babies born >37 weeks.  These are the infants that are substantially more likely to be born through MSAF.  

One of the issues raised by the Chiruvolu article is the increase in the incidence of non-MAS respiratory morbidity following implementation of the no-suction protocol. To stay consistent with this paper, we evaluated infants > 35 weeks gestation at birth and reported non-MAS respiratory morbidity. If the reviewer permits, we would prefer to continue with the same gestational age cohort.

In addition, the VON data indicate that babies with MAS make up approximately 1.5%-1.8% of babies >35 weeks gestation admitted to NICUs.  This is roughly one-third the incidence reported from an even larger population (J Perinatol 2009;29:497-503).  Can the authors comment on this difference? 

The Singh BS et al paper (J Perinatol 2009) reported a MAS incidence of 1.8% of all neonates and 4.6% of term (> 37 weeks) NICU admissions. We reported the incidence of 1.5% to 1.8% among inborn infants at ≥ 35 weeks gestation. The lower gestational age and limitation to inborn births only along with changes in the prevalence of MSAF over the last decade might have accounted for the lower incidence in the VON database.

Additionally, can the authors explain whether or not there are efforts by VON to verify data collected at their 311 centers?  This could be done in a small representative sample of sites as a check for accuracy. Otherwise, do we just have to assume that all the data have been collected and entered correctly?

We did not verify data from a representative sample. Members submit data using standardized definitions and carefully defined eligibility criteria as outlined in the Manual of Operations. Data submitted to Vermont Oxford are subjected to extensive range, logic, and consistency checks.  Each center submitting data completes a data verification plan and attests to its implementation each year before data are merged with the Vermont Oxford Network database.

Of interest in this manuscript, among infants with MAS diagnosed in 2017, there were higher incidences of moderate/severe HIE, surfactant use and inhaled nitric oxide use.  This was true in MAS infants overall, as well as in those with one-minute Apgar scores <3. The authors choose to speculate these findings reflect the elimination of less sick infants who respond to ventilation.  What is worrisome is an alternative hypothesis… the lack of intubation and suctioning has led to sicker infants.  Unfortunately, there is no way to be certain whether either of these conjectures is correct. In addition, the authors speculate that the increased use of nitric oxide and surfactant in 2017 is partly due to increased knowledge subsequent to the 2013-2015 epoch about the latter therapies’ benefits in ameliorating hypoxemic respiratory failure.  This is fallacious, as knowledge about such benefits of these therapies have been known for 15-20 years!

We agree with the reviewer and apologize for the speculation. It is clearly possible that the severity of HIE and PPHN might be increased by reduced tracheal suctioning. We have added a paragraph to outline this concern. We thank the reviewer for highlighting this finding.

The authors should not use a one-minute Apgar score <3 as a proxy for the non-vigorous MSAF infant.  The status of being non-vigorous has been specifically defined as an infant assessed immediately after birth as having a heart rate < 100 and/or poor respiratory effort and/or poor tone.  Moreover, the intubation procedure in MSAF infants is associated with lower one-minute Apgar scores.   Unless the current investigators can reference a study that has specifically correlated the aforementioned definition of non-vigorous MSAF infants with one-minute Apgar scores, they cannot justify this assumption and should eliminate this category.

We do not have a reference to justify an Apgar score of ≤ 3. However, it does represent a subset of infants with perinatal depression. The significant increase in the use of iNO, surfactant and the incidence of moderate/severe HIE in 2017 is an important trend that may have potential clinical significance.

 I applaud the authors acknowledging the limitations to their study and that this precludes their being able to infer that the new guidelines are safe or effective.  Nevertheless, they stated that their findings are “reassuring”, presumably meaning that the 2016 NRP changes have not resulted in worse outcomes. Unfortunately, I believe the limitations in their study prevent them from saying their results are reassuring, particularly in the face of the 2018 publications of Chiruvolu et al. and Singh et al.  I wholeheartedly concur that a large, multicenter, prospective randomized controlled trial is needed to assess tracheal suction in non-vigorous infants born through MSAF.

We agree with the reviewer and have eliminated the sentence referring to the reassuring nature of the results. We thank the reviewer for his/her support for a multicenter trial.

Reviewer 2 Report

Edwards et al present a very robust cohort study comparing the effects on expectant management for non-vigorous meconium stained infants. They used the VON database and found a number of improvements in neonatal outcomes compared to prior years where providers intubated non-vigorous meconium stained newborns. The review is well done I just have a few clarifications.

When citing the three small trials on meconium I think its fair to say the was a trend (p=0.05) in decreasing MAS with intubation versus stating that no trials found a difference. In addition I would also highlight the possibility that some centers are not practicing this change and it would strengthen the validity of the findings if they could send a brief survey as to what year centers changed their practice.

Author Response

Edwards et al present a very robust cohort study comparing the effects on expectant management for non-vigorous meconium stained infants. They used the VON database and found a number of improvements in neonatal outcomes compared to prior years where providers intubated non-vigorous meconium stained newborns. The review is well done I just have a few clarifications.

We thank the reviewer for his/her positive comments.

When citing the three small trials on meconium I think its fair to say the was a trend (p=0.05) in decreasing MAS with intubation versus stating that no trials found a difference.

We have modified this sentence to reflect this trend towards significance.

 In addition I would also highlight the possibility that some centers are not practicing this change and it would strengthen the validity of the findings if they could send a brief survey as to what year centers changed their practice.

We agree that this would be interesting.  However, inconsistency in practices and skills among different providers (as highlighted by Reviewer # 1 within centers would make sampling and interpretation difficult using a survey.  Instead, we are highlighting the need for prospective observational studies to complement the necessary RCTs. 

We thank both reviewers for their constructive criticism.

Round 2

Reviewer 1 Report

The authors have revised their manuscript substantially and addressed most of the concerns I previously brought up.  They have acknowledged the major limitations of their study and have appropriately stated that why cannot use their results to determine that the new guideline is either safe or effective.  Again, my preference would be that they limit their study population to infants of term gestation (37 weeks or greater).  There is a possibility that including large numbers of infants of 35 or 36 weeks gestation may have "diluted" the overall findings of the entire population resulting in a seeming decrease in MAS that may not be present had they limited their study population to those infants of 37 weeks gestation or greater.  Indeed, their median gestational age of 38 weeks in this paper is almost 2 weeks lower than the mean gestational age of Chiruvolu et al's population.  Do Edwards and colleagues have the ability to calculate a mean gestational age so we can better compare with Chiruvolu's work?  I think it is important for Edwards et al. to reference in their Discussion the B.S. Singh paper (J. Perinatology 2009;29:497-503) and state that in a latter manuscript addressing NICU admissions in a more mature population (babies of 37 weeks or greater gestational age), the percentage of admissions due to MAS was considerably higher (4.8%) than what they found.

Author Response

We thank the reviewer for his / her insightful comments. We have addressed the concerns and added the following section to the discussion and referenced the Singh et al paper. Due to time constraints we are not in a position to perform a detailed analysis of term infants but do report the incidence of MAS at term gestation. As the reviewer rightly predicted, the incidence is higher at this gestation.

Changes to the manuscript at line 199-200:

We included infants ≥35 weeks’ GA to be consistent with Chiruvolu et al. Other observational studies used different gestational age criteria and observed higher rates of MAS. Singh et al evaluated a large dataset of 415,772 neonates and reported that 1.8% of all neonates and 4.6% of all term NICU admissions had MAS [15]. Using a gestational age cut-off of ≥ 37 weeks at birth, we observed a decline in the incidence of MAS from 2.3% in 2013-15 to 1.7% in 2017 among all infants and from 8.1% in 2013-15 to 5.4% in 2017 among infants with Apgar score of ≤3 at 1 minute. Temporality, different types of data (administrative vs. clinical), and differences in inclusion criteria (inborn and outborn (transferred) infants vs. inborn infants only) may explain the lower incidence of MAS in our population.